# Using a Traction Table for Fracture Reduction during Minimally Invasive Plate Osteosynthesis (MIPO) of Distal Femoral Fractures Provides Anatomical Alignment

**DOI:** 10.3390/jcm12124044

**Published:** 2023-06-14

**Authors:** Martin Paulsson, Carl Ekholm, Roy Tranberg, Ola Rolfson, Mats Geijer

**Affiliations:** 1Department of Orthopaedics, Sahlgrenska University Hospital, 41345 Gothenburg, Sweden; 2Department of Orthopaedics, Institute of Clinical Sciences, Sahlgrenska Academy, University of Gothenburg, 41345 Gothenburg, Sweden; 3Department of Radiology, Sahlgrenska University Hospital, 41345 Gothenburg, Sweden; 4Department of Radiology, Institute of Clinical Sciences, Sahlgrenska Academy, University of Gothenburg, 41345 Gothenburg, Sweden; 5Department of Clinical Sciences, Lund University, 22185 Lund, Sweden

**Keywords:** femur fracture, orthopedic surgical procedure, fracture osteosynthesis, fracture fixation, internal, minimally invasive surgical procedure, operating table, bone misalignment

## Abstract

Introduction: Fracture reduction and fixation of distal femur fractures are technically demanding. Postoperative malalignment is still commonly reported after minimally invasive plate osteosynthesis (MIPO). We evaluated the postoperative alignment after MIPO using a traction table with a dedicated femoral support. Methods: The study included 32 patients aged 65 years or older with distal femur fractures of all AO/OTA types 32 (c) and 33 (except 33 B3 and C3) and peri-implant fractures with stable implants. Internal fixation was achieved with MIPO using a bridge-plating construct. Bilateral computed tomography (CT) scans of the entire femur were performed postoperatively, and measurements of the uninjured contralateral side defined anatomical alignment. Due to incomplete CT scans or excessively distorted femoral anatomy, seven patients were excluded from analyses. Results: Fracture reduction and fixation on the traction table provided excellent postoperative alignment. Only one of the 25 patients had a rotational malalignment of more than 15° (18°). Conclusions: The surgical setup for MIPO of distal femur fractures on a traction table with a dedicated femoral support facilitated reduction and fixation, resulting in a low rate of postoperative malalignment, despite a high rate of peri-implant fractures, and could be recommended for surgical treatment of distal femur fractures.

## 1. Introduction

Fractures of the distal femur show a bimodal age distribution. One peak comprises high-energy trauma in young males, and the second, more significant, peak comprises elderly females with low-energy trauma [1,2]. Distal femoral fractures are, however, difficult injuries for patients and surgeons alike. It is advantageous to use minimally invasive techniques for obtaining fracture reduction and surgical fixation of distal femoral fractures instead of open surgery. The minimally invasive technique preserves the periosteal blood supply, which is beneficial for fracture healing [3,4,5]. The minimally invasive plate osteosynthesis (MIPO) technique is typically carried out on a standard operating table, with the patient’s leg draped free. However, the lack of clear landmarks in the femur makes per-operative validation of reduction difficult, particularly of rotation, regardless of whether a nail or plate is used, and mal-reduction is not uncommon [4,6,7,8,9,10,11,12]. Postoperative malalignment increases the risk of non-union [13,14]. Rotational malalignment has also been shown to have a negative impact on knee function, resulting in articular cartilage shearing, which can, in turn, develop into painful osteoarthritis [15,16,17].

The use of a traction table for fracture reduction and per-operative fracture immobilization is standard for both proximal and femoral shaft fractures. Using a traction table for distal femur fractures has also been described. However, the reports are scarce; they consist of technical notes and a report of one case with a comminuted distal femur fracture [18,19,20]. The technique of using a traction table equipped with a dedicated femoral support has been employed as the standard surgical setup for operating distal femur fractures in our hospital for more than ten years. However, whether the use of a traction table improves the accuracy of surgical reduction has not previously been reported. 

In this prospective cohort study, we aimed to evaluate to what degree anatomic alignment could be achieved if closed reduction of a distal femur fracture on a traction table with a dedicated femoral support was used when performing a minimally invasive plate osteosynthesis (MIPO) plating and to compare results with previously published findings using a conventional operating table setup. 

## 2. Materials and Methods

### 2.1. Patient Cohort

Patients in this study originate from a randomized controlled trial (RCT) of weight-bearing regimens following fixation of distal femur fracture. All patients were operated with MIPO using a traction table [21]. In the RCT, we included 32 patients aged 65 years or older with a traumatic fracture of the distal part of the femur of 2018 AO/OTA types 33 (A2-3, B1-2, C1-2) and 32(c) (A1-3, B2-3, C2-3) and unified periprosthetic classifications system (UPCS) IV (3B1, 3C-3D) and V (3B1, 3C-3D) [22]. 

For the current study, 25 out of the original 32 eligible patients were included. Three patients were excluded due to incomplete postoperative computed tomography (CT) scans (the entire length of both femurs was not available), and four due to excessively distorted femoral anatomy from previous fractures or surgery, making measurements unreliable.

The mean age was 81 years (standard deviation, SD, 9.0; range 67–95), mean operation time was 96 min (SD 28.0; range 52–175), and the mean blood loss was 272 mL (SD 147.1; range 75–600). None of the commonly seen complications related to the patient setup on the traction table was observed pre- or postoperatively [23]. The distribution of fracture types according to the 2018 AO/OTA classification [22] is shown in Table 1.

Sixteen of the 25 patients (64%) had had previous surgery on the injured femur with either a joint replacement (Table 1) or osteosynthesis (four patients with antegrade intramedullary nail and one with hip screws), and one patient had both a total knee replacement and an antegrade intramedullary nail. 

### 2.2. Surgical Intervention

One of seven predetermined consultant orthopedic trauma surgeons performed the surgical procedure according to a written protocol. The patients were supine on an operating table, with the fractured leg in traction. Bi-planar fluoroscopy was used. An adjustable femoral-supporting device (POsterior Reduction Device P.O.R.D. Orthofix™ SRL, Verona, Italy) mounted on the side of the operating table was used for dorsal support of the fracture (Figure 1). Before skin washing and draping, closed reduction was performed. To reduce the fracture’s commonly occurring apex posterior angulation, the knee was flexed about 20° by lowering the foot stand, keeping the femur horizontal (Figure 2). To further improve the reduction, the height of the femoral support and the amount of traction on the leg could be adjusted. Only mild traction was allowed. The rotation of the leg was set at neutral, with the foot pointing upwards (Figure 3). For a true lateral view with the image intensifier, the foot rotation could be slightly adjusted to align the dorsal contours of the femoral condyles horizontally. However, as specified by the protocol, no further actions to improve the rotational alignment were undertaken.

A small longitudinal incision was made over the lateral epicondyle. The fascia lata was incised longitudinally, and the epicondyle cleared of periosteal tissue. An LCP^®^ Distal Femoral Plate (Synthes™, Oberdorf, Switzerland) was introduced under the fascia lata and fitted to the lateral femoral epicondyle and the femoral diaphysis (Figure 4). A large clamp was used to hold the plate firmly onto the lateral condyle and shaft to achieve best possible contact between the plate and the lateral femoral. When correctly seated, as verified by fluoroscopy, the plate was temporarily fixed with distal and proximal K-wires. The correct plate position is also crucial for optimal distal screw positioning [25,26]. Anatomical reduction of the fracture was prioritized over the approximation of the plate to the femoral proximal shaft (Figure 5 and Figure 6).

A 13-hole plate was used for patients of short stature, while in the other patients, the fractures were fixed with a 15-hole plate. Five bi-cortical locking screws were used in the distal part of the plate, and three bi-cortical locking screws were used in the proximal portion of the plate, the latter through stab incisions. The proximal screws were spaced between 5 holes to distribute the proximal load. The fixation was a bridge-plating construction, and no additional screws or cerclage wires were used across the fracture.

### 2.3. Image Evaluation

For the assessment of angular measurements and measurements of femoral length, all patients underwent a CT scan of both complete femurs within one week after surgery using a metal-artifact reduction reconstruction algorithm and archived as 3-mm contiguous slices. 

Rotation was measured on axial multiplanar reformations (MPR), oriented perpendicular to the long axis of the femur. The rotational angle was defined as the angle between the dorsal condylar line of the distal femur and a line from the apex of the lesser trochanter through the center of the femoral medullary canal (Figure 7). If the patient had a total knee replacement, the corresponding parts of the prosthesis were used for the distal measurement. The vertical side of the image was used as reference.

The length of the femur was measured on a coronal MPR from the most cranial part of the femoral head to the center of the line that connected the most distal contour of the condyles (Figure 8a). CT slice thickness was set to 50 mm to facilitate the determination of the femoral outlines. If the patient had had previous surgery on the proximal part of the femur, the apex of the lesser trochanter was instead used as a measuring reference. 

Genu varum/valgum or coronal angulation was measured using the abovementioned plane. CT slice thickness was set to 50 mm to facilitate the determination of the femoral outlines. The angle between the distal joint line and the mechanical axis of the femur was measured (from the center of the femoral head through the center of the knee) (Figure 8b). With a total hip replacement, the center of the femoral head of the prosthesis was used, even though some total hip replacements did not have an anatomical offset.

On a sagittal MPR plane, perpendicular to the dorsal femoral condylar line, genu antecurvatum or recurvatum (sagittal angulation) was measured as the angle between the longitudinal axis of the distal part of the femoral diaphysis and a line cutting the center of the distal metaphysis at the funnel-shaped transition into the flared portion of the distal shaft (Figure 9). 

### 2.4. Statistical Methods

The analysis was performed in SPSS Statistics, Version 28 (IBM, New York, NY, USA). The normality of continuous data was analyzed with Q-Q plots. Data with a normal distribution are presented as mean and standard deviation (SD) and as median and interquartile range (IQR) if the distribution is non-normal. Intraclass correlation (ICC) was used to calculate the accuracy and intra-rater agreement of the CT scan measurements. Single-rater, multiple measurements (CT scans) on the same femur at different time points gave intra-rater reliability ICC with 2-way mixed effects and an absolute agreement of 95% [27]. 

All measurements were made by the first author (MP) on two occasions, six months apart, using Xero Viewer (web-based software) (AGFA, Mortsel, Belgium). The contralateral femur was used as a reference, and a malalignment score (coronal, sagittal, and length), presented by Handolin et al. [28] was used to evaluate the quality of the postoperative reduction. The mean value of the two independent observations on both femurs is presented. 

## 3. Results

### Reduction Measurements

Only the rotational side-to-side difference had a normal distribution. The mean postoperative difference in the rotation was 5.8° (SD 4.3°, range 18.2° internal rotation to 10.6° external rotation). The distribution is presented in Figure 10. The median length difference was 5.0 mm (IQR 3.0–6.8 mm). One patient had a 10 mm-longer fractured femur due to femoral side-to-side variances (Figure 11) [29]. The median coronal angulation difference (varus or valgus) was 1.2° (IQR 0.4–2.0°), and the median sagittal angulation difference (genu antecurvatum or recurvatum angulation) was 0.8° (IQR 0.4–1.2°). Using the threshold values for malalignment suggested by Handolin et al. [28], all patients were categorized as “excellent”. 

The results of the intra-observer agreement of the reduction measurements assessed by ICC are shown in Table 2. 

## 4. Discussion

The results of this study show that, in a cohort of 25 elderly patients with distal femur fractures surgically treated with closed reduction on a traction table followed by MIPO fixation, “excellent” alignment was achieved in all patients, according to the Handolin et al. malalignment score [28]. Rotational malalignment of more than 15° was seen in only one patient, which is less frequent than previously published results [8,9,10]. We believe these results are clinically relevant, as improved postoperative alignment correlates with improved functional outcomes and decreased risk of non-union [13,14]. Improved rotational alignment also lowers the risk of articular cartilage shearing, negatively affecting knee function and promoting osteoarthritis [15,16,17]. 

Despite the advances in modern surgical techniques, postoperative malalignment is still being reported. In a recent multi-center RCT comparing nail and plate fixation in 126 distal femur fractures by Dunbar et al. [12], postoperative reduction was one of the outcomes assessed. The overall coronal postoperative malalignment (>5°) after MIPO fixation was found to be 32%. Valgus deformity was more common than varus deformity, at 27.4% and 4.8%, respectively, but no sagittal malalignment was found. Sagittal malalignment was, however, reported to be common in comminute periprosthetic distal femur fractures in a report by Sharma et al. [11]. In a report on pitfalls when applying lateral plates in distal femoral fractures, Collinge et al. [26] describes strategies for reducing displaced distal femur fractures with varus/valgus angulation and ante-/recurvatum angulation using fluoroscopy on a standard supine operating table. Rotational malalignment, however, is more challenging to assess and manage [26,30]. Buckley et al. [8], using CT, found a side-to-side rotational difference of 22.3–31.3° in 3 of 13 patients treated with MIPO fixation for a distal femur fracture. In a cohort study on rotational alignment of both distal femur fractures (38 patients) and femoral diaphyseal fractures (13 patients) after MIPO (on a traditional operating table), Kim et al. [9] concluded that, while the coronal and sagittal alignments were satisfactory in 96% of the patients, only 57% had satisfactory rotational alignment, using 8° as a threshold. However, the distribution of malrotation is not reported, and the results are, therefore, not presented in Table 3. Furthermore, Kim et al. referred to the grading of malrotation according to the Handolin et al. score. Still, the Handolin et al. score does not include a rotational malalignment component, as all scoring was performed on plain X-rays [28]. Lill et al. [10] concluded that the MIPO technique yielded significantly higher degrees of rotational malalignment (5 of 10 patients with >15°), typically external rotation, than open reduction and internal fixation (surgical setup not specified) using magnetic resonance imaging (MRI, Table 3). 

Compared to these previous studies, the traction table method used in the present study did provide an excellent reduction in all patients, according to the Handolin et al. criteria [28], and a better rotational alignment (Table 3).

To evaluate the result of the reduction, the contralateral unfractured femur is used as a reference, but the natural side-to-side rotational differences make its use somewhat unreliable [29,31,32]. Sutter et al. [33] compared unfractured femurs in 63 individuals, primarily investigating anatomical differences as a potential cause of hip symptoms using MRI. They found a significant average side-to-side difference of 4° of the femoral neck anteversion. Croom et al. [34] investigated both unfractured femurs in 164 patients with CT. Eighteen percent had an anteversion difference of over 10°, and 4% had over 15°. There is currently no evidence to specify a clinically relevant rotational malalignment threshold. However, Croom et al. suggested that the threshold for clinically relevant rotational malalignment should be 15°, considering natural side differences. Applied to the patients in the present study, only one of the 25 patients was fixed with remaining rotational malalignment.

Using a traction table has several advantages that could tribute to the beneficial outcome in the present study [20,35]. The traction table is the most commonly used setup for proximal and femoral shaft fractures; therefore, most orthopedic surgeons are familiar with its use. The setup allows for gross reduction without being hindered by draping or affecting sterility. Traction to reduce distal femur fractures is not a novel invention; traction was the most commonly used treatment before the era of open reduction and fixation with osteosynthesis. The treatment with traction until healing produced acceptable results [36]. By applying traction to the fractured leg, the reduction mechanism is caused by increasing the tension of the soft tissue (periosteum, muscles, and tendons). The ligamentotaxis self-aligns and, thereby, reduces the fracture [8]. Lowering the foot stand, combined with the dedicated dorsal support, neutralizes the pulling force of the gastrocnemius muscles and prevents apex-posterior malalignment (Figure 2). Sixteen of the 25 patients in the present study had a spiral fracture in the distal part of the shaft, fractures considered unstable and often with a rotational displacement [37]. The setup on the traction table, with the foot in a neutral forward position, is usually sufficient to reduce the fracture when traction is applied. Furthermore, the use of the traction table eliminates the need for the time-consuming visual recognition of rotational malalignment reference points [25], such as a bilateral frontal plane view of the lesser trochanter, which has been recommended to assess malrotation [30]. Lastly, the achieved reduction is maintained throughout the surgery. This eliminates the need for an assistant to maintain manual traction for reduction during the surgery. It also diminishes the need for an invasively applied external fixator or distractor.

The use of the MIPO technique in the fixation of distal femur fractures has been shown to be advantageous, with less violation of blood supply at the fracture site and, subsequently, a reported decrease of non-unions [3,4,38,39,40,41,42,43,44]. Obtaining a maximal contact area between the femoral bone surface and the plate reduces stress at the bone-implant interface. It reduces the risk of failure, especially in osteoporotic bone [45,46]. Using a long bridging plate with locking screws has also been shown to have biomechanical advantages [47,48,49,50], such as the lowest incidence of loss of fixation, more flexibility, and better capability to withstand permanent deformation in osteoporotic bone, compared to other fixation options [51,52,53,54,55,56]. The bridging plate concept used in the current study has also been reported to decrease the risk of non-unions [57,58,59,60]. 

There are, however, also limitations to the present study. Using the contralateral leg as a reference is standard practice, despite the inherent side-to-side variations. Thus, the contralateral leg may not accurately compare with the fractured femur. Differences may be amplified when the patients have had previous surgery, such as osteosynthesis or joint replacements. In addition, measurements of osteoporotic bone can be difficult. 

Measurements of the sagittal alignment were challenging, reflected by the relatively low ICC. The combination of metal artifacts from total knee replacement, the proximity of screws from osteosynthesis, osteoporotic bone, and 3-mm thick CT slices did not allow angle measurements using Blumensaat’s line for reference [61]. The method used for measuring sagittal angulation in the present study has not been validated, and the results should be interpreted with that in mind. However, the technique used for sagittal angulation measurements is rarely described in previous reports [9,11,28]. For measurements of the rotational alignment, we used the lesser trochanter and the center of the femoral medullary canal as the proximal reference. The main reason for choosing the lesser trochanter for reference was the high frequency of hip implants (total hip replacements and osteosynthesis). Previous studies have used the center of the femoral head, femoral neck, and greater trochanter, although they lie in different planes [8,9,10]. To our knowledge, using the lesser trochanter as a reference has not yet been validated.

The patients in the current study are typical for the elderly cohort who are most likely to sustain a distal femur fracture [1,2]. Although there were only 25 patients included in this study, which is slightly more than in most previous studies (Table 3), the limited number of patients could influence the external validity. Ideally, a larger cohort of patients would contribute to knowledge and external validity by comparing the two different surgical setups, traction table vs. traditional operating table.

One apparent strength is that the study is prospective. All the measurements were performed on full-length femur CT scans obtained within a week of operation before potential secondary displacement could occur. The reliability of the measurements showed a high ICC in rotation, length, and coronal plane but lower in the sagittal plane due to the lack of clear anatomic structures for measuring references in the sagittal plane. 

## 5. Conclusions

The surgical setup technique for distal femur fractures with MIPO on a traction table with a dedicated femoral support provided an “excellent” result in all patients, according to the malalignment score by Handolin et al. [28]. Using the threshold for malrotation suggested by Croom et al. (15° side-to-side difference) [34], only 1/25 (18°) had malrotation, which is a lower rate of malrotation than previously published results on MIPO for distal femur fracture. The assessed surgical setup was easy to use and proved to be a valuable tool in the challenging task of reducing and fixating a distal femur fracture with MIPO. Further research with larger sample sizes comparing surgical results depending on a traditional operating table setup vs. a traction table setup would provide valuable knowledge on this topic.

## Figures and Tables

**Figure 1 jcm-12-04044-f001:**
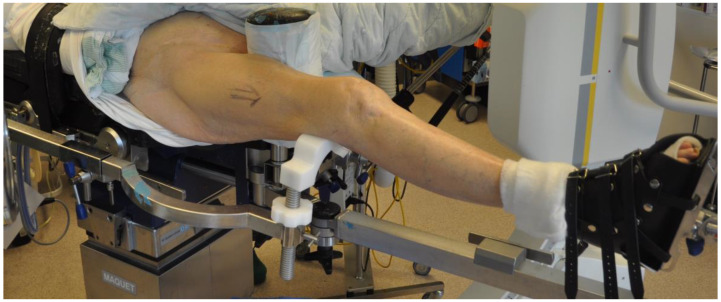
The patient on the traction table with the leg in traction. A femoral support (POsterior Reduction Device P.O.R.D. Orthofix™ SRL, Verona, Italy) [24] is used to reduce the commonly occurring apex posterior angulation of the distal femoral fracture (red arrow).

**Figure 2 jcm-12-04044-f002:**
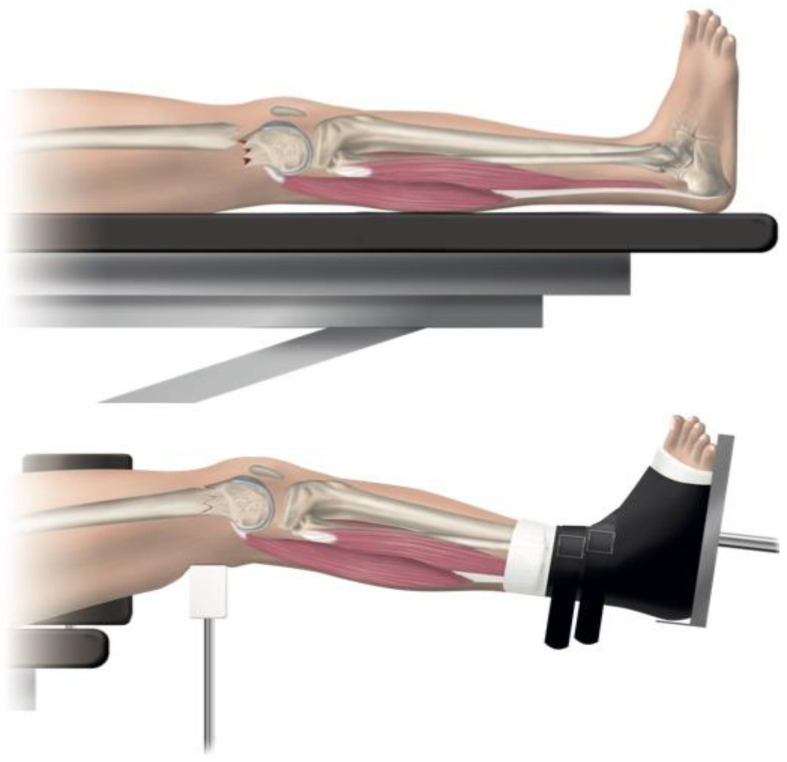
(**Top**) Apex posterior angulation of the distal femoral fracture, which is commonly occurring. The pull of the gastrocnemius muscles attached to the femoral condyles causes the typical displacement pattern. (**Bottom**) With the patient on the traction table, closed reduction was obtained by gentle traction and lowering the foot stand, while the fracture was supported from dorsal by the dorsal femoral support. The reduction could be further improved by adjusting the height of the femoral support, the foot stand, and the amount of traction on the leg.

**Figure 3 jcm-12-04044-f003:**
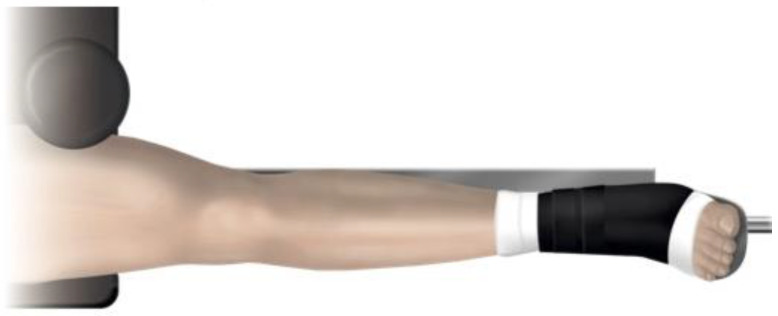
The patient on the traction table with the leg in traction and the foot pointing upwards (seen from above), which reduces preoperative coronal or rotational malalignment. The patient’s position on the table is important, as the central post can affect the angulation of the fracture in the frontal plane.

**Figure 4 jcm-12-04044-f004:**
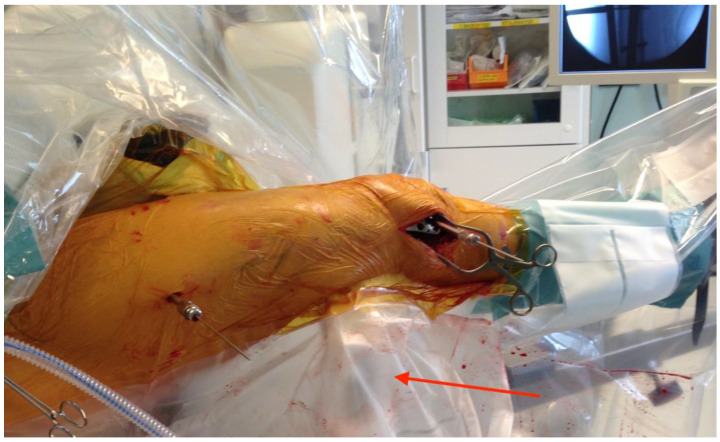
The patient setup on the operating table, with the fractured leg in mild traction and fracture supported by an adjustable femoral-supporting device. The red arrow points towards the femoral support, partly hidden behind the draping. A small incision has been made over the lateral condyle. The fascia lata has been longitudinally incised, and an LCP^®^ Distal Femoral Plate (Synthes™, Oberdorf, Switzerland) has been introduced under the fascia lata and fitted to the lateral femoral epicondyle and the femoral diaphysis. The plate is temporarily fixed with distal and proximal K-wires.

**Figure 5 jcm-12-04044-f005:**
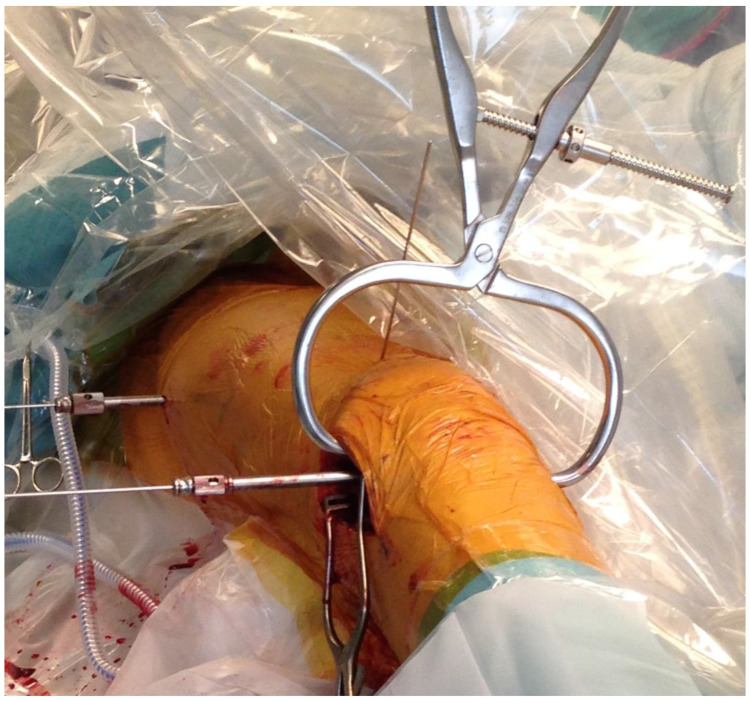
The plate was firmly pressed onto the lateral condyle by a large clamp to achieve as much contact as possible between the plate and the femoral surface.

**Figure 6 jcm-12-04044-f006:**
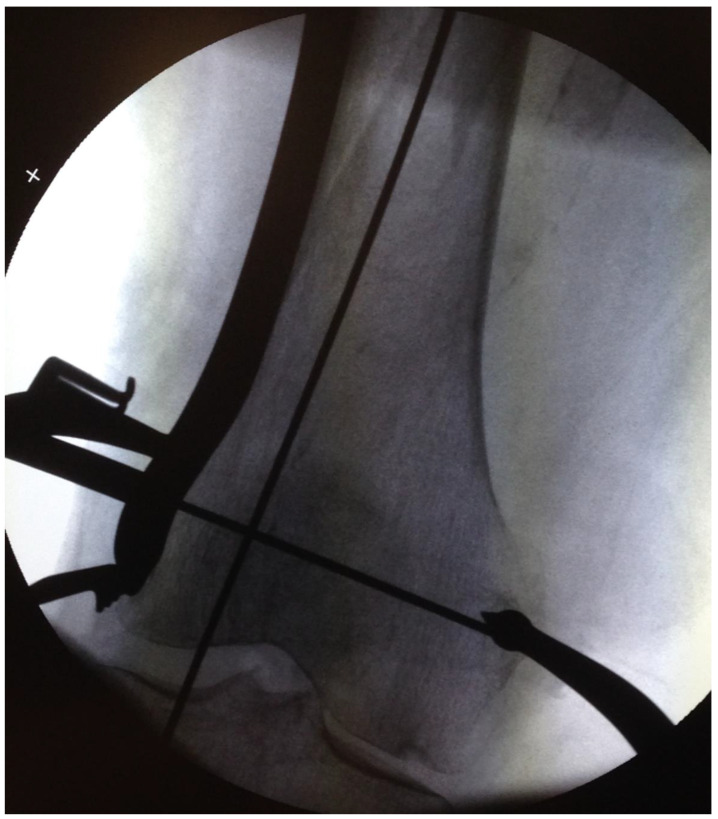
Image from the bi-planar image intensifier. A large clamp firmly pressed the plate onto the lateral condyle and diaphysis. The plate was temporarily fixed with K-wires. The spiral-shaped fracture of the distal shaft was reduced by a combination of traction, a dedicated femoral support, and the application of an LCP^®^ Distal Femoral Plate (Synthes™, Oberdorf, Switzerland), firmly pressed onto the lateral epicondyle.

**Figure 7 jcm-12-04044-f007:**
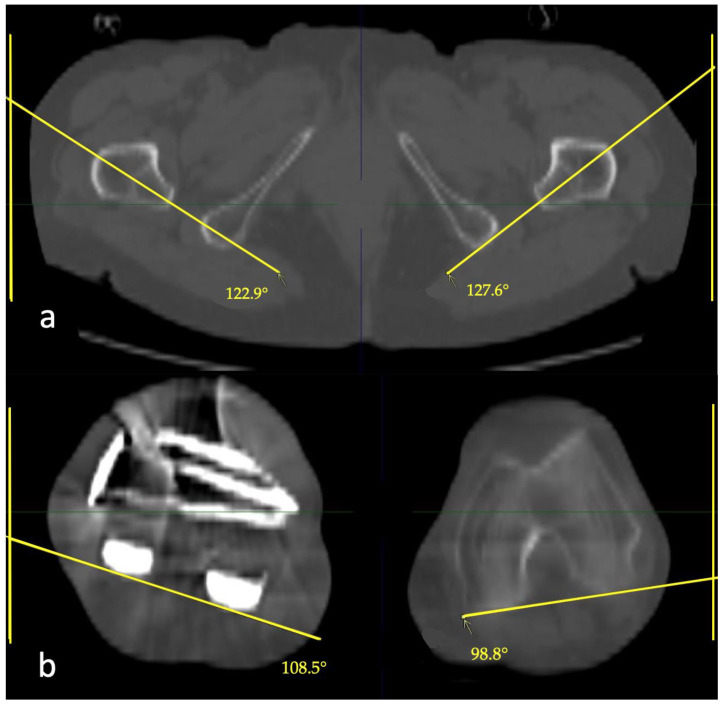
(**a**) Proximal rotational angles were measured by a line passing through the apex of the lesser trochanter and the center of the femoral medullary canal. (**b**) Distal rotational angles were measured using the dorsal condylar line. The vertical side of the image was used for reference.

**Figure 8 jcm-12-04044-f008:**
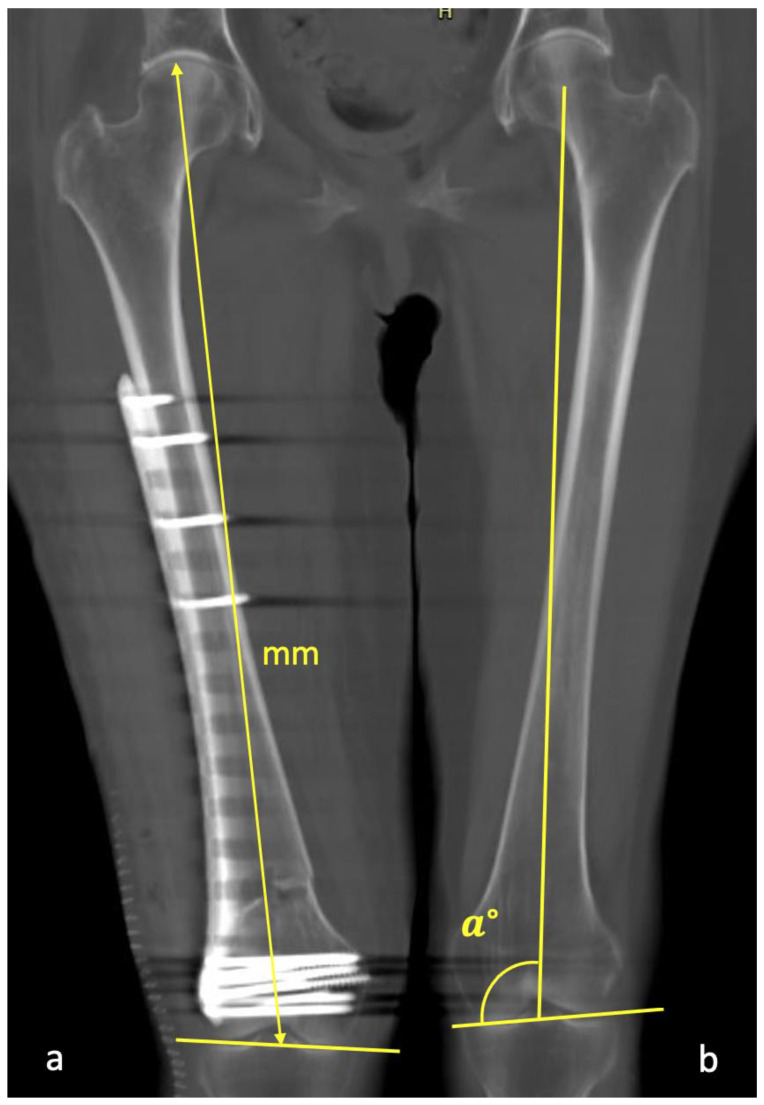
(**a**) The total femoral length was measured from the most cranial part of the femoral head to the center of the line that connected the most distal contour of the medial and lateral condyles. (**b**)The varus/valgus angulation was measured between the joint line using the distal contour of the femoral condyles and the mechanical axis of the femur from the center of the femoral head through the center of the knee.

**Figure 9 jcm-12-04044-f009:**
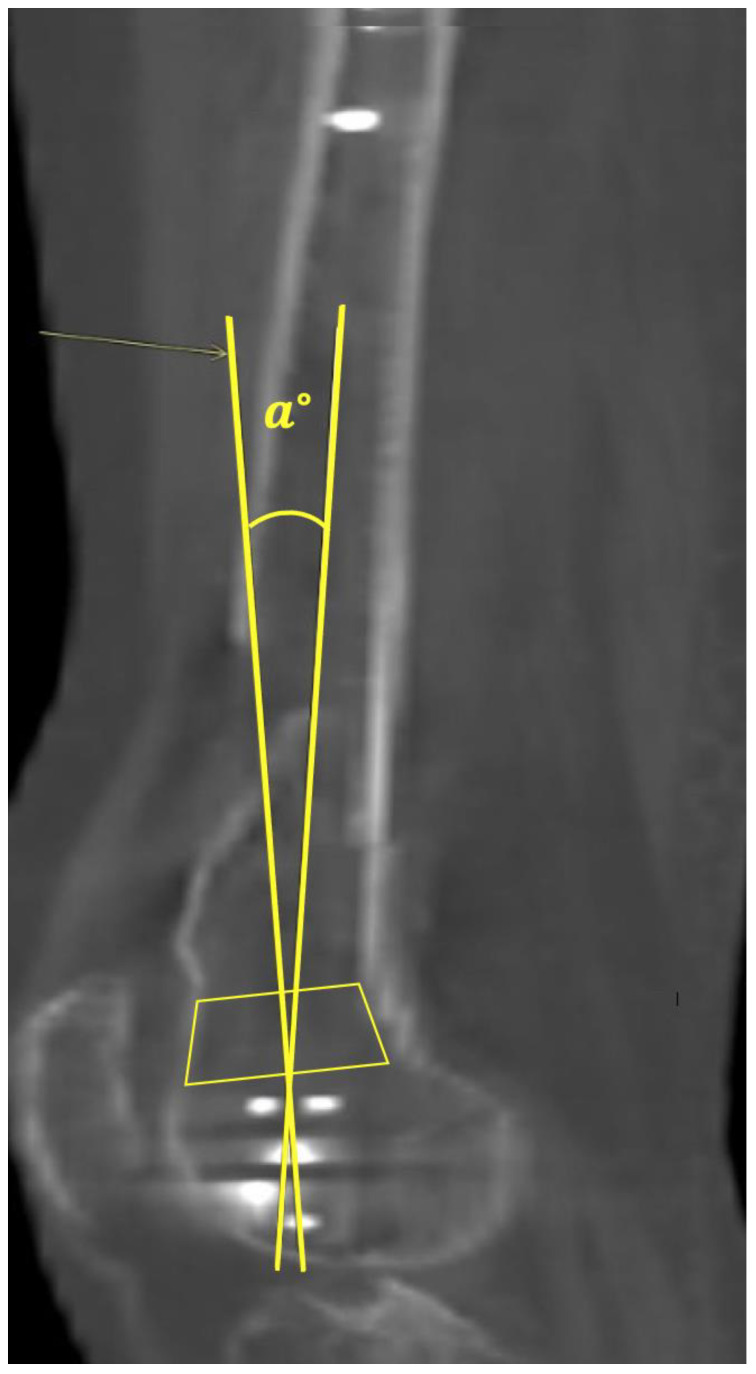
Genu antecurvatum/recurvatum (sagittal angulation) was defined as the angle between a line passing through the center of the distal diaphysis and a line cutting the center of the distal metaphysis at the funnel-shaped transition into the flared shaft on a sagittal multiplanar reformation.

**Figure 10 jcm-12-04044-f010:**
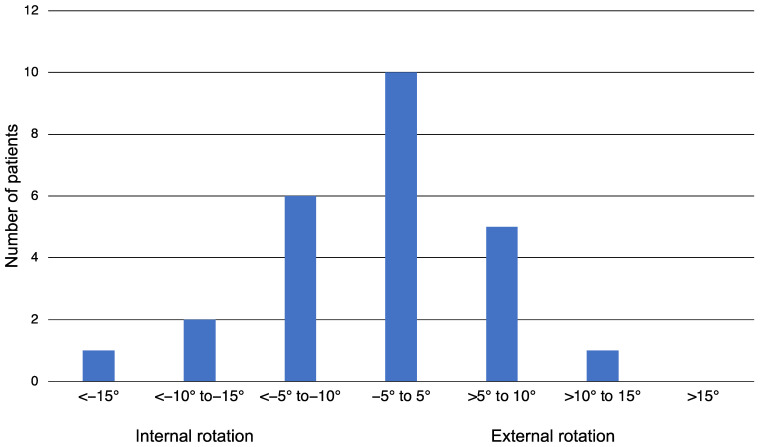
The diagram shows the distribution of postoperative rotational side-to-side differences in the investigated patients. The negative values represent an internal rotation compared with the unfractured femur, and the positive values represent external rotation.

**Figure 11 jcm-12-04044-f011:**
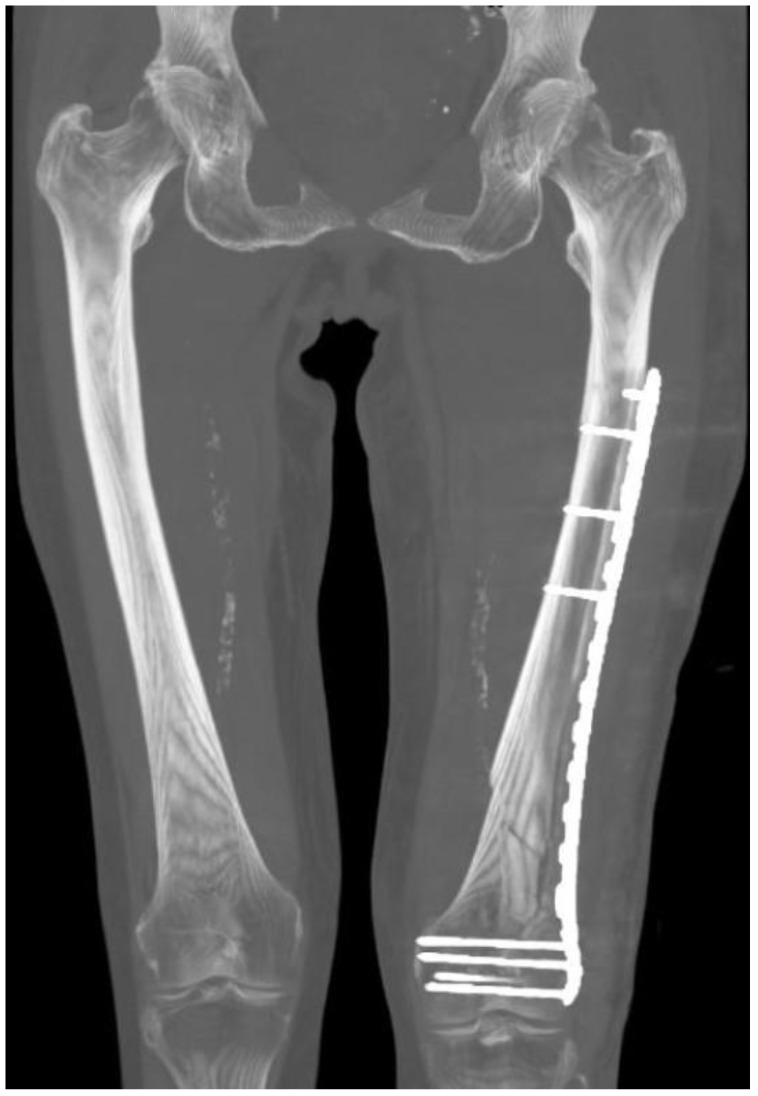
An image of a computed tomography scan of both femurs. The image was obtained by setting the slice thickness to 50 mm. This patient had a longer femur on the fractured side and no history of previous injuries. The fracture was reduced and fixated without elongation. Side-to-side variances in femoral shape are not uncommon [29].

**Table 1 jcm-12-04044-t001:** AO/OTA Classification.

n	AO/OTA	UM	Q	UCPF
2	32A2.1	(c)		
4	32A2.1	(c)	[12]	
1	32A2.1	(c)		V.3C
1	32A2.1	(c)	[12]	V.3C
2	32B2	(c)	[13]	
2	32B2	(c)	[13]	V.3D
2	32B3	(c)	[13]	
1	32B3	(c)	[13]	IV.3C
1	32B3	(c)	[13]	IV.3C
1	33A2.2			V.3B1
2	33A3.2			V.3B1
1	33A3.2			
1	33A3.2		[7]	V.3B1
1	33A3.2		[7]	IV.3D
1	33B2.1			
1	33C1.1			IV.3C
1	33C2.3			

AO/OTA; Arbeitsgemeinschaft für Osteosynthesefragen/Orthopaedic Trauma Association, UM; Universal Modifiers, Q; Qualifications, UCPF; Unified Classification System for Periprosthetic Fractures.

**Table 2 jcm-12-04044-t002:** Intraclass correlation, postoperative computed tomography (CT) scans TEST-RETEST.

ROTATION	n	ICC	95% CI
Distal Angle Fracture	25	0.985	(0.964−0.993)
Distal Angle Non-fracture	25	0.951	(0.877−0.975)
Proximal Angle Fracture	25	0.919	(0.823−0.963)
Proximal Angle Non-fracture	25	0.975	(0.823−0.963)
GENU VARUM/VALGUM			
Fracture	25	0.959	(0.909−0.982)
Non-fracture	25	0.968	(0.928−0.986)
LENGTH			
Fracture	25	0.999	(0.998−1.000)
Non-fracture	25	0.999	(0.998−1.000)
GENU ANTE-/RECURVATUM			
Fracture	25	0.596	(0.271−0.800)
Non-fracture	25	0.544	(0.198−0.770)

CI; confidence interval, Fracture; the fractured femur. Non-fracture; the unfractured femur.

**Table 3 jcm-12-04044-t003:** Studies on malrotation after surgery for distal femoral fractures using MIPO—minimally invasive plate osteosynthesis.

	Modality	Surgical Intervention	n	Mean Age (Min–Max)	Proportion RM 10–15°	Proportion RM > 15°
Buckley et al., 2011 [8]	CT	MIPO	13	38.1	15%	23%
Lill et al., 2016 [10]	MRI	MIPO + ORIF	10 + 10	44.8 (17–91)	20% + 20%	50% + 0%
Current study	CT	MIPO	25	81.4 (67–95)	12%	4%

CT; Computed Tomography, MRI; Magnetic Rensonace Imaging, RM; Rotational Malalignment, ORIF; Open Reduction Internal Fixation.

## Data Availability

The data presented in this study are available on request from the corresponding author. The data are not publicly available due to national legislation.

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
