# Peer review of "Using a Traction Table for Fracture Reduction during Minimally Invasive Plate Osteosynthesis (MIPO) of Distal Femoral Fractures Provides Anatomical Alignment"

_jcm, 2023, doi:10.3390/jcm12124044_

Round 1

Reviewer 1 Report

The article titled "Postoperative Alignment in Distal Femur Fractures Using MIPO and Traction Table" focuses on evaluating the efficacy of using a traction table with a dedicated femoral support during minimally invasive plate osteosynthesis (MIPO) for distal femur fractures (DFFs). The authors aim to address the common issue of postoperative rotational malalignment associated with MIPO and determine if the use of a traction table can improve postoperative alignment outcomes.

The study included 32 patients aged 65 years or older with DFFs, excluding specific fracture types. MIPO with a bridge-plating construct was used for internal fixation. Postoperatively, bilateral computed tomography (CT) scans of the entire femur were performed to assess the alignment by comparing measurements with the uninjured contralateral side. Seven patients were excluded due to incomplete CT scans or excessively distorted femoral anatomy.

The results of the study indicate that the utilization of a traction table with a dedicated femoral support during MIPO for DFFs resulted in excellent postoperative alignment. Out of the 25 patients analyzed, only one patient exhibited rotational malalignment greater than 15° (18°). This finding suggests that the surgical setup involving a traction table facilitated successful fracture reduction and fixation, resulting in a low rate of postoperative malalignment despite the high prevalence of peri-implant fractures.

The authors' conclusion highlights the potential benefits of employing a traction table with a dedicated femoral support during MIPO for DFFs. This approach appears to enhance the reduction and fixation of fractures, leading to improved postoperative alignment outcomes. The study implies that this surgical setup can be recommended as a viable treatment option for DFFs.

However, it is important to note certain limitations of the study. The relatively small sample size of 25 patients after excluding seven individuals might affect the generalizability of the findings. Additionally, the study primarily focused on postoperative alignment outcomes and did not extensively explore other important clinical parameters such as functional outcomes or complications associated with the procedure.

In summary, the study provides valuable insights into the use of a traction table with a dedicated femoral support in MIPO for DFFs. The findings suggest that this surgical setup can potentially improve postoperative alignment outcomes and reduce rotational malalignment. Further research with larger sample sizes and a comprehensive assessment of clinical outcomes is warranted to validate these findings and elucidate the broader impact of this technique on patient outcomes.

Author Response

Comments and Suggestions for Authors

The article titled "Postoperative Alignment in Distal Femur Fractures Using MIPO and Traction Table" focuses on evaluating the efficacy of using a traction table with a dedicated femoral support during minimally invasive plate osteosynthesis (MIPO) for distal femur fractures (DFFs). The authors aim to address the common issue of postoperative rotational malalignment associated with MIPO and determine if the use of a traction table can improve postoperative alignment outcomes.

The study included 32 patients aged 65 years or older with DFFs, excluding specific fracture types. MIPO with a bridge-plating construct was used for internal fixation. Postoperatively, bilateral computed tomography (CT) scans of the entire femur were performed to assess the alignment by comparing measurements with the uninjured contralateral side. Seven patients were excluded due to incomplete CT scans or excessively distorted femoral anatomy.

The results of the study indicate that the utilization of a traction table with a dedicated femoral support during MIPO for DFFs resulted in excellent postoperative alignment. Out of the 25 patients analyzed, only one patient exhibited rotational malalignment greater than 15° (18°). This finding suggests that the surgical setup involving a traction table facilitated successful fracture reduction and fixation, resulting in a low rate of postoperative malalignment despite the high prevalence of peri-implant fractures.

The authors' conclusion highlights the potential benefits of employing a traction table with a dedicated femoral support during MIPO for DFFs. This approach appears to enhance the reduction and fixation of fractures, leading to improved postoperative alignment outcomes. The study implies that this surgical setup can be recommended as a viable treatment option for DFFs.

However, it is important to note certain limitations of the study. The relatively small sample size of 25 patients after excluding seven individuals might affect the generalizability of the findings. Additionally, the study primarily focused on postoperative alignment outcomes and did not extensively explore other important clinical parameters such as functional outcomes or complications associated with the procedure.

In summary, the study provides valuable insights into the use of a traction table with a dedicated femoral support in MIPO for DFFs. The findings suggest that this surgical setup can potentially improve postoperative alignment outcomes and reduce rotational malalignment. Further research with larger sample sizes and a comprehensive assessment of clinical outcomes is warranted to validate these findings and elucidate the broader impact of this technique on patient outcomes.

Submission Date

29 April 2023

Date of this review

23 May 2023 10:25:27

Dear Reviewer 1;

Thank you for taking the time to review our article. Your comments and suggestions are appreciated and highly valued. Please be advised that the significant increase in manuscript text and figures is on behalf of the strong recommendations from the JCM to have a minimum of 4000 words, references excluded.

  1. The introduction has been partly rewritten and extended, the background has been improved, and we have added a couple of references.
  2. Thank you for pointing out the need to reflect on the generalizability of the findings. We have clarified this limitation in the manuscript in the limitation section. The patients included in this study were also included in an RCT [1], where the functional outcome was evaluated. As described in the methods, no complications related to the surgical setup were seen in this study.
  3. Since this study is the first to report on the results of closed reduction and MIPO on a traction table, we agree that larger sample size studies are needed to validate the findings of this study and have added this to the manuscript in the limitations and conclusion.

  1. Paulsson M; Ekholm C; Jonsson E; Geijer M; Rolfson O. Immediate full weight-bearing versus partial weight-bearing after plate fixation of distal femur fractures in elderly patients. A randomized controlled trial. Geriatric Orthopaedic Surgery & Rehabilitation 2021, 12, doi:doi.org/10.1177/21514593211055889.

Reviewer 2 Report

In this prospective cohort study, it aimed to evaluate to what degree anatomic alignment could be achieved if closed reduction of a DFF on a traction table with a dedicated femoral support was used when performing a minimally invasive plate osteosynthesis (MIPO) plating and to compare results with previously published findings using a conventional operating table setup. The authors give comments that should be addressed as follows.

1.      Line 56, the authors stated computed tomography (CT). Please give the explanation regarding the basic concept and explanation of CT. Also, refer relevant reference as follow, doi: 10.3390/ma16093298, 10.3390/su15010823, and 10.3390/biomedicines11020427

2.      Line 96, why should use LCP® Distal Femoral Plate? Is there the product is the best in market? Or having any advantages?

3.      Line 101, the procedure of image evaluation would be improved with providing illustrative image.

4.      Line 140, the basis of statistical methos should be explained.

5.      Line 170, in figure 7. Please explain the trend shows from the results.

6.      In the present form, actually nothing really novel. The current works appears to be a replication or modified literature according to the lack of novelty. The authors must extensively describe the novel in their work. This work should be rejected due to a serious concern.

7.      In order to highlight the gaps in the literature that the most recent literature aims to fill, it is crucial to review the benefits, novelty, and limitations of earlier studies in the introduction.

8.      Line 201 in table 3, why should adopt/shows thee reflected research? Any explanation to adopt it? Is collected from literature systematic searching?

-

Author Response

Comments and Suggestions for Authors

In this prospective cohort study, it aimed to evaluate to what degree anatomic alignment could be achieved if closed reduction of a DFF on a traction table with a dedicated femoral support was used when performing a minimally invasive plate osteosynthesis (MIPO) plating and to compare results with previously published findings using a conventional operating table setup. The authors give comments that should be addressed as follows.

Dear Reviewer 2;

Thank you for taking the time to review our article. Your comments and suggestions are appreciated and highly valued. Please be advised that the significant increase in manuscript text and figures is on behalf of the strong recommendations from the JCM to have a minimum of 4000 words, references excluded.

  1. Line 56, the authors stated computed tomography (CT). Please give the explanation regarding the basic concept and explanation of CT. Also, refer relevant reference as follow, doi: 10.3390/ma16093298, 10.3390/su15010823, and 10.3390/biomedicines11020427

Thank you, we have clarified why the three patients were excluded due to incomplete CT scans. Thank you also for providing three very interesting articles on porous materials and Finite Element Analysis. The analyzing technique used in the articles is interesting but, unfortunately, not applicable in the evaluation of fracture reduction in this study.  

  1. Line 96, why should use LCP® Distal Femoral Plate? Is there the product is the best in market? Or having any advantages?

This is a legitimate question. The reasons for choosing the LCP plate are multiple; At the timepoint of initiating and planning this study, the LCP distal femoral plate was the only anatomical distal femur plate available in our hospital. The LCP distal femoral plate has been one of the most used plates for distal femur fractures in the last twenty-five years. No other plate has been scrutinised as extensively. It would be fair to say that the LCP is the most validated implant for distal femur fractures. Newer implants may prove superior in the future, but no study so far has shown better results than the LCP to our knowledge.

  1. Line 101, the procedure of image evaluation would be improved with providing illustrative image.

Thank you; We agree that the description of the procedure for measuring the different angles and lengths of the femur can be considered complicated. However, are there clarifications of the measurement procedures on the CT images in the text and in the legends of  Fig 7-9.

  1. Line 140, the basis of statistical methos should be explained.

Good point; Since this study is observational, there was no statistical comparison between groups or patients.

  1. Line 170, in figure 7. Please explain the trend shows from the results.

Thank you for pointing this out; The diagram shows the distribution of the different degrees of malrotation in the evaluated patients. We have clarified this in the figure legend.

  1. In the present form, actually nothing really novel. The current works appears to be a replication or modified literature according to the lack of novelty. The authors must extensively describe the novel in their work. This work should be rejected due to a serious concern.

Thank you; We have made some improvements to the introduction and discussion, where we clarify that the surgical setup evaluated in this study is not novel; it is used for hip fractures and femoral shaft fractures, but its standardized use for distal femur fractures has not been described. Regarding the literature on plate fixation of distal femur fractures using closed reduction on a traction table, it has to our knowledge, only been described in a couple of technical notes and in one description of a severely comminuted distal femur fracture. However, the postoperative results of the reduction using this setup have never been evaluated previously, which could be considered more important as it seems to improve postoperatively alignment more than the conventional setup.

  1. In order to highlight the gaps in the literature that the most recent literature aims to fill, it is crucial to review the benefits, novelty, and limitations of earlier studies in the introduction.

Good point; We have made some improvements to the introduction and the discussion to highlight the need for improved postoperative alignment. We also emphasize that there are no previous studies made on evaluating postoperative alignment using a traction table. We have also clarified some limitations of methods used in previous studies on postoperative reduction using MIPO with a traditional surgical setup.

  1. Line 201 in table 3, why should adopt/shows thee reflected research? Any explanation to adopt it? Is collected from literature systematic searching?

Good question; There are, to our knowledge, no available systematic reviews on postoperative alignment on distal femur fractures after MIPO. There are currently three papers on this topic, to our knowledge. Two of these papers are presented both in the discussion and in Table 3. In the third paper, Kim et al. (not included in Table 3, but discussed in the discussion), the results are not reported in a way that they can be used in the table.

Comments on the Quality of English Language

-

Submission Date

29 April 2023

Date of this review

09 May 2023 06:20:38

Round 2

Reviewer 2 Report

Reviewers greatly appreciate the efforts that have been made by the author to improve the quality of their articles after peer review. I reread the author's manuscript and further reviewed the changes made along with the responses from previous reviewers' comments. Unfortunately, the authors failed to make some of the substantial improvements they should have made making this article not of decent quality with biased, not cutting-edge updates on the research topic outlined. In addition, the author also failed to address the previous reviewer's comments, especially on comments number 1 (not incorporated the literature) and 6 (lack of novel). Thank you very much for the opportunity to read the author's current work. 

-

Author Response

Dear reviewer.

Thank you for taking the time to read and comment on the revised manuscript.

We apricate the feedback on the first revision of the manuscript; however, we find it difficult to understand how to improve the manuscript from the comments given in the current review.

We will, however, try to clarify the issues regarding the comments on questions/comments 1 and 6 from reviewer 2 first round;

  1. Line 56, the authors stated computed tomography (CT). Please give the explanation regarding the basic concept and explanation of CT. Also, refer relevant reference as follow, doi: 10.3390/ma16093298, 10.3390/su15010823, and 10.3390/biomedicines11020427

We have difficulties understanding this request, we have used the modality of computed tomography scans, which is a standard radiological modality. One could, of course, argue that MRI would provide less radiological exposure, but MRI does have the disadvantage of being more sensitive to metal artefacts. The patient had plain x-rays of the fractured leg taken at the same time. Still, the use of CT scans is more accurate and provides a calculated reference of length which is not provided by the x-ray. Additionally, rotational angulation can not be measured on plain x-rays. Furthermore, do we not understand the intention of the suggested papers. As stated in the first response to this comment, we do not understand in what way they are related to this study.

We have, however, added a sentence in the methods on why CT scans were utilised.

  1. In the present form, actually nothing really novel. The current works appears to be a replication or modified literature according to the lack of novelty. The authors must extensively describe the novel in their work. This work should be rejected due to a serious concern.

This comment, which appears more to be a statement, we find difficult to comment on.

We have clarified in the manuscript that there are currently only three reports in the literature on using the traction table for distal femur fractures. In two of these ( instructions on surgical techniques in distal femur fractures), using the traction table is mentioned as an alternative to the standard operating table, and in one recent technical note on its use in complicated distal femoral fractures. We have made changes to the manuscript where we have clearly stated that this method has been used and described previously in the literature. Still, it has not yet been evaluated in a clinical study.